# Triple Notches Bandstop Microstrip Filter Based on Archimedean Spiral Electromagnetic Bandgap Structure

**Xuemei Zheng [1,2] and Tao Jiang [1,\*]**

[1] College of Information and Communication Engineering, Harbin Engineering University, Harbin 150001, China
[2] College of Electrical Engineering, Northeast Electric Power University, Jilin 132012, China
\* Correspondence: jiangtao@hrbeu.edu.cn; Tel.: +86-451-8251-9808

**Abstract:** With the development of artificial electromagnetic structures, defective grounding structures (DGS), defective microstrip structures (DMS), and electromagnetic bandgap (EBG) have been widely used in the design of microstrip filters. In this paper, a triple notches ultra wideband bandstop microstrip filter based on Archimedean spiral electromagnetic bandgap structure (ASEBG) structure is proposed. Firstly, the equivalent circuit of ASEBG is analyzed, and L and C values are extracted by using Advanced Design System (ADS). Secondly, the correctness of the lumped parameter model is verified by comparing the High Frequency Structure Simulator (HFSS) simulation results with the measured results. Finally, the influence of ASEBG structure parameters on resonant performance is analyzed by HFSS simulation, and the filter parameters are further optimized. By coupling ASEBG structure to existing double notch microstrip filters, a triple notches ultra wideband bandstop microstrip filter is realized. This method can also be used in the design of other microstrip devices with stopband characteristics. The three bandgap center frequencies of the proposed triple notches ultra wideband bandstop microstrip filter are 3.5, 5.2, and 7.4 GHz, respectively. The corresponding maximum attenuation of the three stopbands is 33.6, 24.8, and 21.7 dB, respectively.

**Keywords:** microstrip notch filter; Archimedean spiral; electromagnetic bandgap structure

---

## 1. Introduction

With the development of artificial electromagnetic structures such as electromagnetic bandgap structure and defective microstrip structure, which are widely used in the design of microstrip antennas, microstrip filters, microstrip couplers, and other RF devices. The ultra wide band (UWB) approved by the Federal Communications Commission (FCC) is 3.1–10.6 GHz [1]. It covers narrowband communication systems such as WIMAX band (3.3–3.7 GHz), WLAN band (5.15–5.35 GHz and 5.725–5.825 GHz) and C band (7.25–7.75 GHz). In order to filter the interference of a multi-narrowband system whose transmitting power is obviously higher than UWB, the design of multi-notch microstrip band-stop filter has become a research focus. Generally speaking, artificial electromagnetic structures generally include defective ground structure (DGS), defective microstrip structure (DMS), electromagnetic bandgap structure (EBG) and so on. So far, scholars have applied artificial electromagnetic structures to the design of microstrip filters. The traditional methods to realize notch microstrip filters such as using open or short stub loaded resonators [2,3]; using stepped impedance resonators [4,5]; using multimode resonators [6–8] such as T shaped resonators [9–11], E shaped resonators [12,13], ring resonators [14], Hilbert-Fork Resonators [15]; coupling resonators are designed by coupling structure between microstrips [16,17]. In addition, microstrip notch filters can

be realized by constructing a defective grounding structure (DGS) [18,19]. The defective grounding structure changes the distribution inductance and capacitance of the transmission line by etching the structure on the grounding metal plate, and obtains band-stop and slow-wave characteristics, but it leaks floor energy. In order to solve the problem of floor energy leakage, a defective microstrip structure is proposed. [20,21], which can also generate slow wave characteristics. In [22], the authors proposed a microstrip notch filter based on electromagnetic bandgap structure, which realized a high rejection broadband filter. At present, the realization of UWB tri-notch band-stop filters mainly focuses on the following aspects: a stub loaded resonator [23], square ring short stub loaded resonators [24], a stepped impedance resonator (SIR) [25], a coupled-line sub-loaded shorted stepped impedance resonator (SIR) [26], multiple resonant and defected ground structure [27], hexagonal metamaterials split ring resonators [28], the wave cancellation technique [29], using cascaded and multi-armed methods [30], using controlled coupling of open-loop-ring defected ground structure [31], using the U-resonator and suspended multilayer-technique [32], using the multilayer technique and coupled octagonal defected ground structure [33]. Most methods for designing triple notches bandstop microstrip filters mentioned above are based on a single resonant structure, and the 3 dB bandwidth of the stopband are narrow for filtering narrow-band interference.

In this paper, a triple notches bandstop microstrip filter based on Archimedean spiral EBG structure and existing dual notch microstrip filter is presented. By analyzing the lumped circuit model of Archimedean spiral EBG structure, the lumped parameter model of triple notches bandstop microstrip filter coupled with Archimedean spiral EBG structure is analyzed. The validity of the model is further verified by comparing the actual measurement results of the filter with the simulation results by using Advanced Design System (ADS) software (Agilent Technologies Inc., KSanta Clara, CA, USA) and High Frequency Structure Simulator (HFSS) software (ANSYS, Inc., Canonsburg, PA, USA). Without changing the existing dual notch filter structure, the proposed filter achieves the third stop band by coupling the ASEBG structure on the realized dual notch filter [9].

## 2. Performance Analysis of Archimedean Spiral EBG Structure

### 2.1. Structural Dimensions

Figure 1a,b shows the top and side views of mushroom electromagnetic band gap (MEBG) structure, respectively. The equivalent circuit diagram of MEBG is shown in Figure 1c. There is a potential difference between the electromagnetic bandgap structure and the microstrip line and the contact floor. $C_0$ is the capacitance between the feeder and the EBG patch. $C_1$ is the capacitance between the EBG patch and the ground plate. $L_1$ is the inductance value on the metal hole.

$$L_1 = \frac{\mu_0}{2\pi}\left[h \cdot \ln\left(\frac{h + \sqrt{r^2 + h^2}}{r}\right) + \frac{3}{2}(r - \sqrt{r^2 + h^2})\right], \tag{1}$$

$$C_1 = \frac{W(\varepsilon_0 \varepsilon_r + \varepsilon_0)}{\pi} \cosh^{-1}\left(\frac{W + d}{d}\right), \quad C_0 = \varepsilon_r \varepsilon_0 \frac{W^2}{h}, \tag{2}$$

$$f = \frac{1}{2\pi \sqrt{L_1 C_1}}, \tag{3}$$

$$Z_S = \frac{1}{j\omega C_1 + \frac{1}{j\omega L_1}} = \frac{j\omega L_1}{1 - \omega^2 L_1 C_1}. \tag{4}$$

In the above equations, $f$ is the resonant frequency, which is determined by capacitance and inductance. $h$ is the thickness of dielectric substrate. $r$ is the metal hole radius of electromagnetic bandgap structure. $W$ is the edge length of electromagnetic bandgap structure patch. $d$ is the gap between electromagnetic bandgap structure and feeder. $\varepsilon_0$ is the dielectric constant of vacuum medium

substrate; $\varepsilon_r$ is the relative dielectric constant of dielectric substrate; The inductance mainly depends on $h$ and $\mu_r$, and the capacitance mainly depends on the side length $W$ and the spacing $d$.

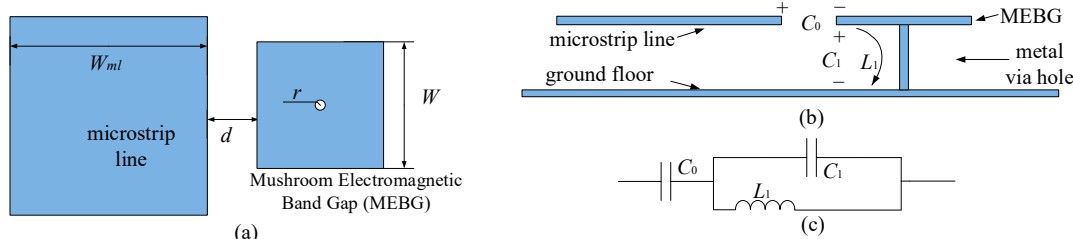

**Figure 1.** (**a**) The top view of the MEBG structure; (**b**) the side view of the MEBG structure; (**c**) the equivalent circuit diagram of the MEBG structure.

The size of traditional MEBG resonator is too large. Adding Archimedean spiral inductance to the surface of the EBG unit can reduce the size of the EBG structure. Archimedean spiral equation is as follows:

$$X(t) = (r + at)\cos(t), \tag{5}$$

$$Y(t) = (r + at)\sin(t), \tag{6}$$

$$t = end\_t - start\_t. \tag{7}$$

In which, $X(t)$ is the transverse coordinate of Archimedean spiral; $Y(t)$ is the axis coordinate of Archimedean spiral; t is the extreme angle, the unit is the degree, indicating the total number of degrees of Archimedean spiral rotation; $r$ is the radius, at $t = 0°$, the unit is mm; $a$ is Archimedean spiral coefficient, the unit is mm/°, which indicates the increase (or decrease) of polar diameter per 1° rotation $start\_t$ is the starting angle of polar angle; $end\_t$ is the maximum angle of polar angle.

Figure 2a,b are the top and side views of Archimedean spiral electromagnetic bandgap structure (ASEBG), respectively, in which $dr$ represent the difference between the inner and outer diameters of Archimedean spiral inductors. The equivalent circuit diagram of ASEBG structure and equivalent circuit for calculating resonance frequency are shown in Figure 2c,d. A new inductance $L_2$ is introduced by loading Archimedean spiral on the surface of mushroom-type EBG patch, which is equivalent to the inductance produced by adding Archimedean spiral to the equivalent circuit of traditional mushroom-type EBG.

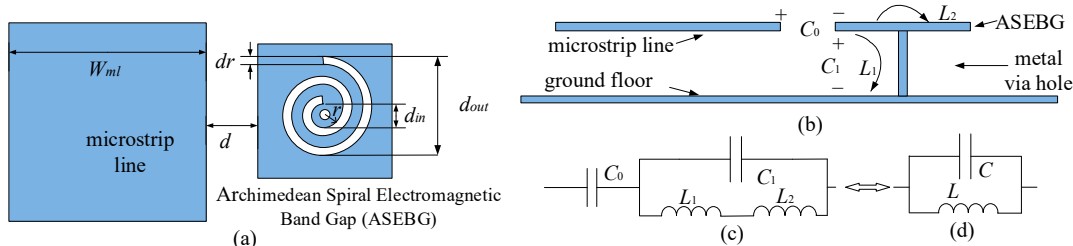

**Figure 2.** (**a**) The top view of ASEBG structure; (**b**) the side view of ASEBG structure; (**c**) the equivalent circuit diagram of ASEBG structure; (**d**) the equivalent circuit for calculating resonance frequency of ASEBG structure.

According to the Wheeler equation [34], the theoretical formula of Archimedean spiral parameters is as follows:

The capacitance between the microstrip and the EBG patch is $C_0$. The capacitance between the EBG structure and the microstrip line and the contact floor is $C_1$. The inductance value on the metal

through hole is $L_1$. The inductance produced by helix in EBG structure is $L_2$. $N$ is the number of coils of inductance. $K_1$ and $K_2$ are the coefficients related to layout, and $d_{avg}$ is the average of $d_{in}$ and $d_{out}$.

$$L_2 = m_0 N^2 \frac{K_1 d_{avg}}{1 + K_2 r}, \tag{8}$$

$$f = \frac{1}{2\pi \sqrt{LC_1}}, L = L_1 + L_2, \tag{9}$$

$$Z_S = \frac{1}{j\omega C_1 + \frac{1}{j\omega L}} = \frac{j\omega L}{1 - \omega^2 LC_1}, \tag{10}$$

$$d_{avg} = (d_{in} + d_{out})/2, (d_{in} = 2r, \quad d_{out} = r + ta), \tag{11}$$

$$r = (d_{out} - d_{in})/(d_{in} + d_{out}). \tag{12}$$

In which, $\varepsilon_0$ is the dielectric constant of the vacuum medium substrate. $\varepsilon_r$ is the relative dielectric constant of the dielectric substrate. $h$ is the thickness of dielectric substrate; $f$ is the resonant frequency (i.e., the stopband frequency center). $r_1$ is the radius of metal through-hole of electromagnetic bandgap structure. $W$ is the edge length of the EBG patch. $d$ is the electromagnetic bandgap structure and feeder spacing. According to Equations (8)–(12), inductance $L_1$ mainly depends on $h$ and $r_1$. Capacitance $C_1$ is mainly determined by side length $W$ and spacing $d$. The resonant frequency center $f$ is determined by capacitance $C_1$ and inductance $L_1$.

As shown in Figure 2c, Archimedean spiral EBG structure is equivalent to adding spiral equivalent inductance $L_2$ to the equivalent circuit, making $L$ larger than mushroom EBG of the same shape. According to Equation (6), the resonance frequency $f$ is inversely proportional to it. Therefore, the resonant frequency of Archimedean spiral EBG structure is lower than that of traditional mushroom EBG, thus realizing the miniaturization of electromagnetic bandgap structure.

### 2.2. Analysis of Resonance Performance of Archimedean Spiral EBG Structure

It is necessary to analyze the effect of different parameters $d$, $W$, $r$, $dr$, $start\_t$, $end\_t$ on resonance characteristics of Archimedean spiral EBG structure by using HFSS. The Archimedean spiral EBG structure is modeled by HFSS. The medium of microstrip line is RT/Duorid 5880, the width of microstrip line is $W_{ml} = 3$ mm, the edge length of EBG structure square is $W = 4$ mm, the distance between EBG and microstrip line is $d = 0.1$ mm, and the radius of metal through hole of Archimedean spiral EBG structure is $radius = 0.3$ mm. $start\_t = 0$ rad, $end\_t = 9$ rad, a = 0.06 mm/°. The sweep frequency analysis of the parameters is carried out by using HFSS, and other parameters are kept unchanged when one parameter is analyzed. The following is an analysis of the simulation results.

#### 2.2.1. The Spacing $d$ between Archimedean Spiral EBG Structure Unit and Microstrip Line

Simulation results with different $d$ by using HFSS are shown in Figure 3a. Figure 3b shows the curves of resonant frequency, maximum attenuation, and 3 dB fractional bandwidth with varying $d$. The simulation results show that $d$ increases from 0.06 to 0.2 mm, the resonant frequency of EBG structure increases from 4.95 to 5.35 GHz, the fractional bandwidth of 3 dB decreases from 6.5% to 2.8%, and the maximum stopband attenuation decreases from 27.33 to 15.27 dB. With the increase of $d$, the 3 dB fractional bandwidth decreases and the maximum attenuation of stopband decreases, and the resonance intensity of EBG resonator decreases. According to Equations (5) and (6), with the increase of $d$, the capacitance $C_0$ decreases, and the resonance frequency $f$ is inversely proportional to $\sqrt{C}$, so $f$ decreases with the increase of $d$.

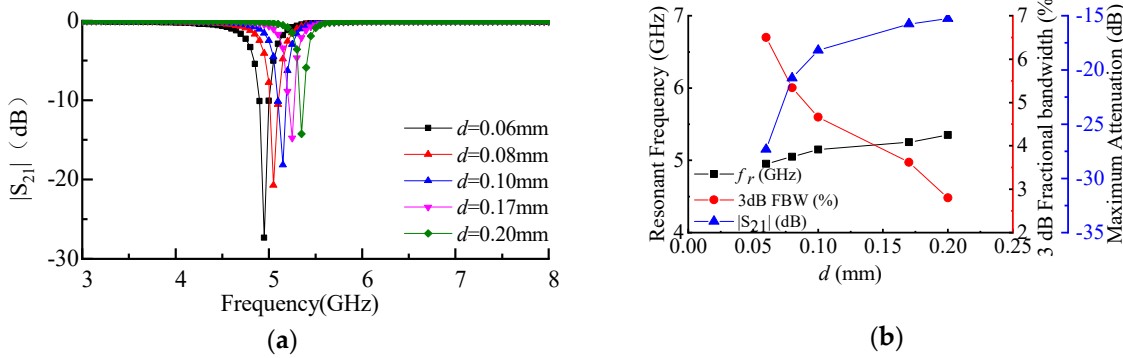

**Figure 3.** (**a**) Simulation results with different *d* by using HFSS. (**b**) Resonant frequency, maximum attenuation, and 3 dB Fractional Bandwidth with different *d*.

### 2.2.2. Edge Length *W* of Archimedean Spiral EBG Structure

Figure 4a shows simulation results with different *W* by using HFSS. Figure 4b shows the curves of resonant frequency, maximum attenuation, and 3 dB fractional bandwidth varying with time. As shown in Figure 4b, with the increase of *W* from 3.4 to 5.2 mm, the resonant frequency *f* decreases from 6 to 4.3 GHz. According to Equation (2), the capacitance $C_1$ and *C* increase with the increase of *W*, and while the resonance frequency *f* is inversely proportional to $\sqrt{C}$. Therefore, the resonant frequency *f* decreases with the increase of *W*.

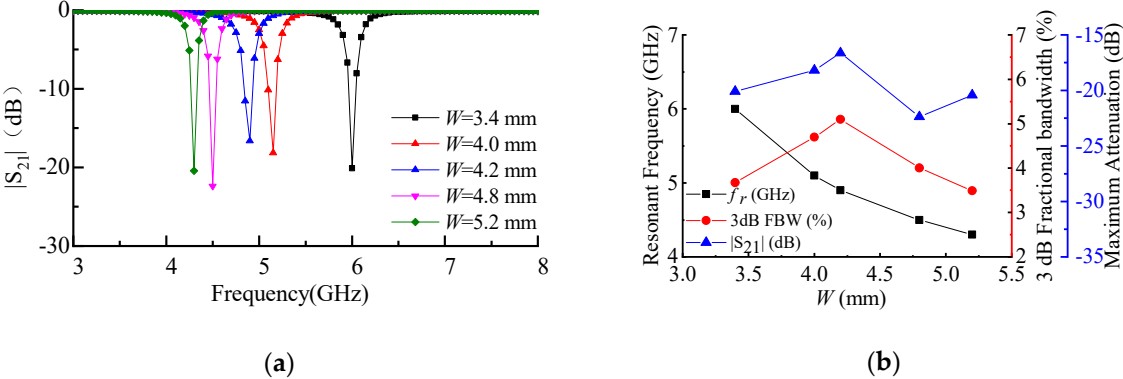

**Figure 4.** (**a**) Simulation results with different *W* by using HFSS. (**b**) Resonant frequency, maximum attenuation, and 3 dB Fractional Bandwidth with different *W*.

### 2.2.3. Polar Diameter *r* of the Archimedean Spiral with at *Start_t = 0* rad

Figure 5a shows the simulation results with different *r* by using HFSS. Figure 5b shows the curves of resonant frequency, maximum attenuation and 3 dB fractional bandwidth varying with different *r*. The results show that the resonant frequency *f* decreases from 5.45 to 4.85 GHz, and the maximum stopband attenuation increases from 17.64 to 23.76 dB with the increasing *r* from 0.24 to 0.42 mm. Equations (8) and (9) show that with the increase of *r*, inductance $L_2$ and *L* increases and resonance frequency *f* decreases for inverse ratio with $\sqrt{L}$.

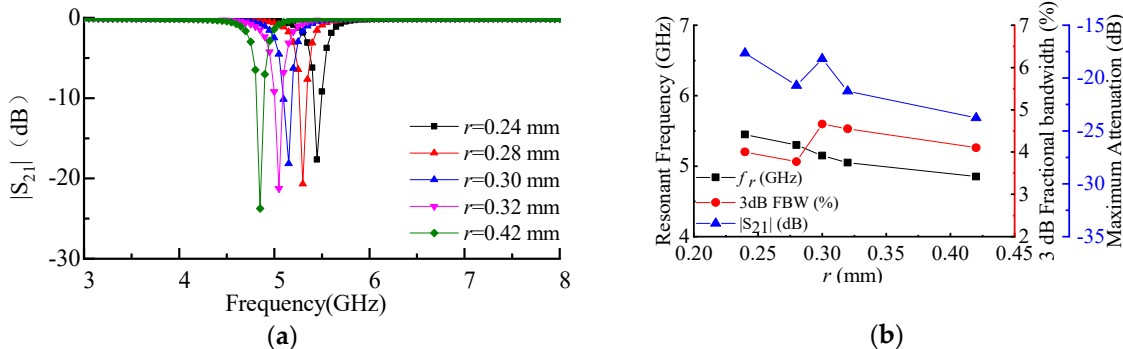

**Figure 5.** (**a**) Simulation results with different *r* by using HFSS. (**b**) Resonant frequency, maximum attenuation, and 3 dB Fractional Bandwidth with different *r*.

### 2.2.4. The Difference of Inner and Outer Diameters of Archimedean Spiral *dr*

Figure 6a shows the simulation results with different dr by using HFSS. Figure 6b shows the curves of resonant frequency, maximum attenuation, and 3 dB fractional bandwidth with varying *dr*. The simulation results show that the resonant frequency f decreases from 6.15 to 4.6 GHz, 3 dB fractional bandwidth decreases from 4.23% to 3.55% and the maximum stopband attenuation increases from 19.07 to 25.28 dB with the increase of dr from 0.1 to 0.28 mm. Figure 5b shows that with the increase of *dr*, the resonant frequency decreases, and 3 dB fractional bandwidth tends to narrow, but the change trend is slower.

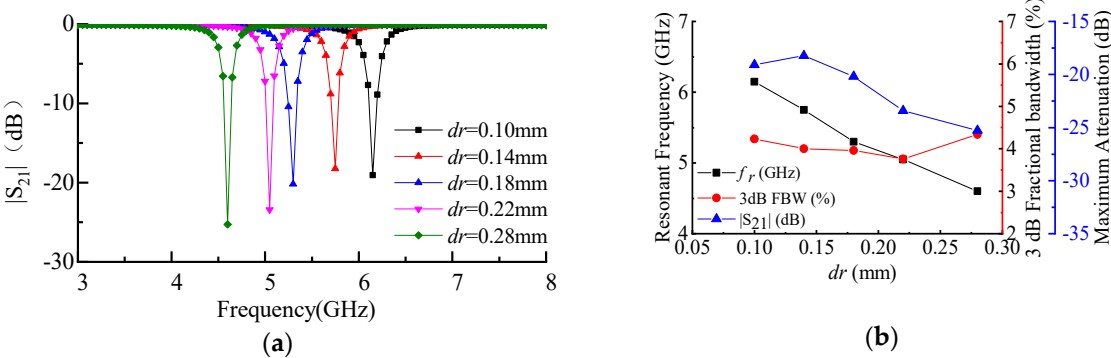

**Figure 6.** (**a**) Simulation results with different *dr* by using HFSS. (**b**) Resonant frequency, maximum attenuation, and 3 dB Fractional Bandwidth with different *dr*.

### 2.2.5. Initial Pole Angle Value of Archimedean Spiral Inductor *Start_t*

Figure 7a shows the simulation results with different *start_t* by using HFSS. Figure 7b shows the curves of resonant frequency, maximum attenuation, and 3 dB fractional bandwidth with different *start_t*. The simulation results show that when *start_t* increases from 0.2 to 2.2 rad, the resonant frequency *f* increases from 5.25 to 6.95 GHz, the 3 dB Fractional Bandwidth increases from 4.01% to 4.89%, and the maximum stopband attenuation decreases from 23.76 to 16.22 dB. Equations (8) and (9) show that with the increase of *start_t* the number of spiral inductor coils decreases, which makes $L_2$ and $L$ decrease. In addition, because the resonance frequency *f* is inversely proportional to $\sqrt{L}$, it increases with the increase of *start_t*.

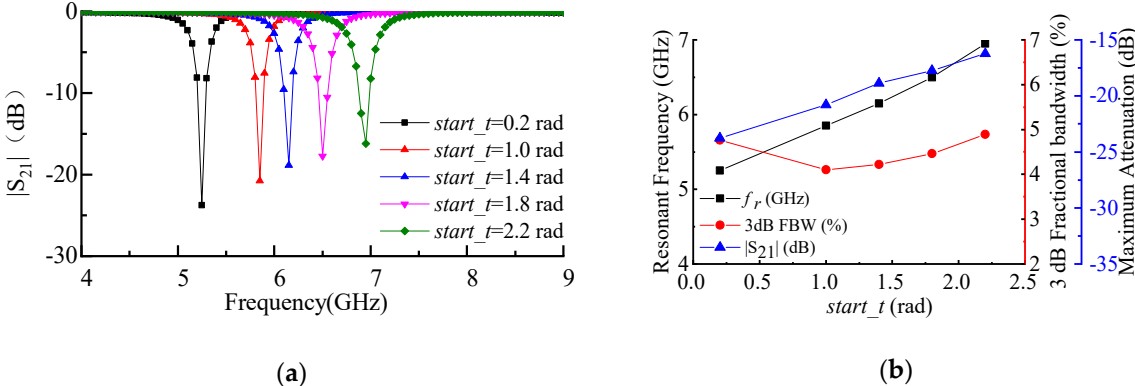

**Figure 7.** (**a**) Simulation results with different *start_t* by using HFSS. (**b**) Resonant frequency, maximum attenuation, and 3 dB Fractional Bandwidth with different *start_t*.

2.2.6. Maximum Pole Angle of Archimedean Spiral Inductor *end_t*

Figure 8a shows the simulation results with different *end_t* by using HFSS. Figure 7b show the resonant frequency, maximum attenuation, and 3 dB Fractional Bandwidth with different *end_t*.

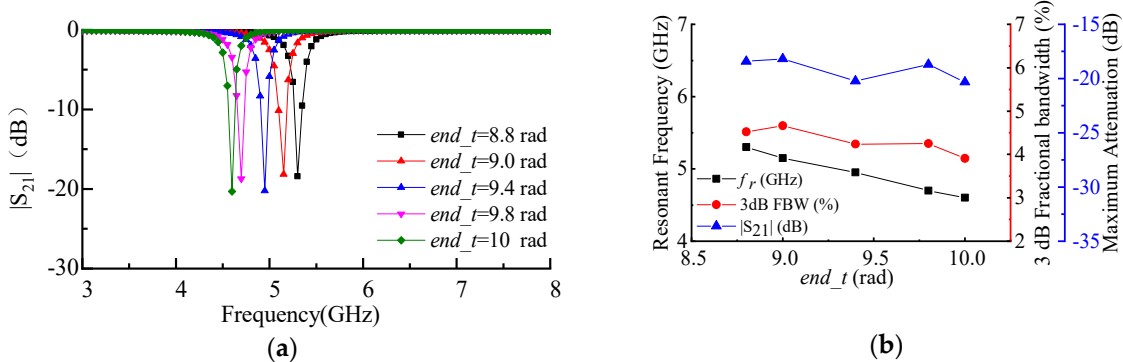

**Figure 8.** (**a**) Simulation results with different *end_t* by using HFSS. (**b**) Resonant frequency, maximum attenuation, and 3 dB Fractional Bandwidth with different *end_t*.

The simulation results show that the *end_t* increases from 8.8 to 10 rad, the resonant frequency of EBG structure decreases from 5.3 to 4.6 GHz, and the fractional bandwidth of 3 dB increases from 4.52% to 3.91%. Equations (8) and (9) show that the increase of *end_t* leads to the increase of spiral inductor coils, while $L_2$ and $L$ increase with the increase of *end_t*. As the resonance frequency f is inversely proportional to $\sqrt{L}$, it decreases with the increase of *end_t*. It can be seen that the resonance characteristics of Archimedean spiral EBG structure vary with the parameters *d*, *W*, *r*, *dr*, *start_t*, and *end_t*. Therefore, the stopband resonance frequency, 3 dB bandwidth, and stopband attenuation of Archimedean spiral EBG structure can be adjusted by adjusting them.

## 3. Filter Design

Based on the previous work of dual notch bandpass filter [9], the ultra-wideband tri notch bandpass filter is realized by coupling Archimedean spiral EBG structure on the dual notch bandpass filter [9]. The design process of the tri notch bandstop filter is shown in Figure 9. Archimedean spiral EBG structure resonator frequency is designed to operate at 5.2 GHz. The structure of triple notches bandstop microstrip filter using MEBG is shown in Figure 10a. The simulation results of the proposed filter by using HFSS are shown in Figure 10b. The structure of the designed triple notches bandstop microstrip filter is shown in Figure 11a. The simulation results of the proposed filter by using HFSS are shown in Figure 11b. The simulation results show that the 3 dB bandwidth of BPF is 3.01~10.79 GHz

and fractional bandwidth is 113.4%. The simulation results of three stopbands of the proposed filter are shown in Table 1.

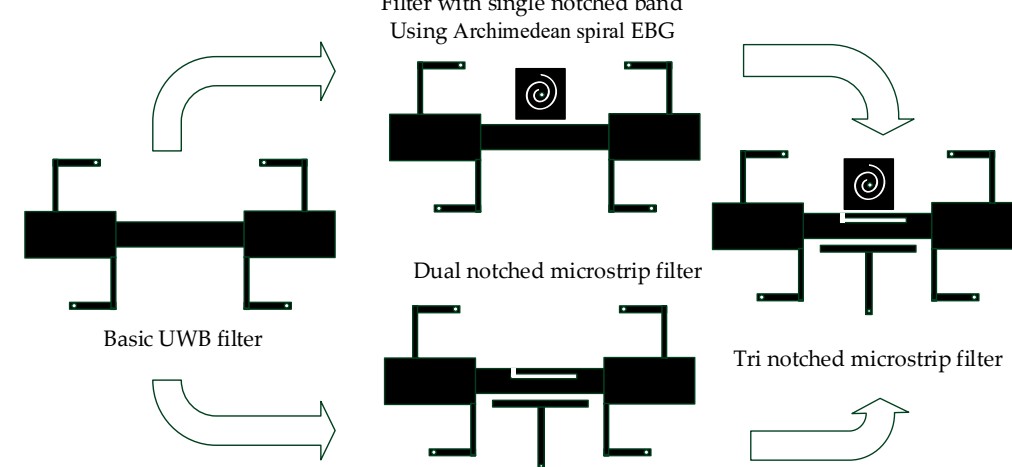

**Figure 9.** The design process of the triple notches bandstop microstrip filter.

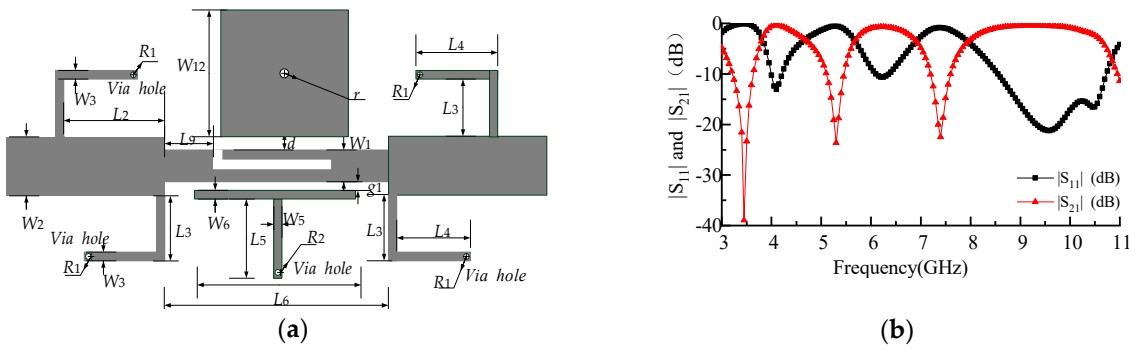

(**a**)　　　　　　　　　　　　　　　　　　　　　　　　(**b**)

**Figure 10.** (**a**) The circuit of the triple notches bandstop microstrip filter using MEBG; (**b**) the simulation results of the proposed filter by using HFSS.

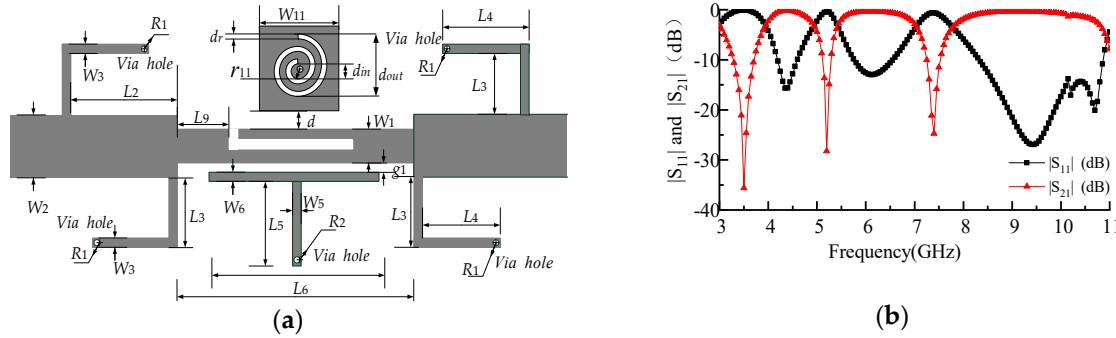

(**a**)　　　　　　　　　　　　　　　　　　　　　　　　(**b**)

**Figure 11.** (**a**) The circuit of the triple notches bandstop microstrip filter using Archimedean spiral electromagnetic bandgap structure (ASEBG); (**b**) the simulation results of the proposed filter by using HFSS.

**Table 1.** Simulation results of the proposed filter by using HFSS.

| Notch Frequency (GHz) | 3 dB Bandwidth (GHz) | 3 dB Fractional Bandwidths (FBW) % | Rejection Level (dB) |
|---|---|---|---|
| 3.5 | 1.07 | 15.62 | 35.61 |
| 5.2 | 0.41 | 7.81 | 28.23 |
| 7.4 | 0.61 | 8.91 | 24.90 |

Figure 12a is the equivalent circuit of Archimedean spiral EBG structure resonant unit. Figure 12b is the equivalent circuit of the dual notch bandpass filter [9]. Figure 12c is the equivalent circuit of the proposed triple notches bandstop microstrip filter. The values of *L* and *C* of the equivalent circuit are calculated from Equation (9). The Lumped circuit is simulated and analyzed, and the values of *L* and *C* are optimized by using ADS.

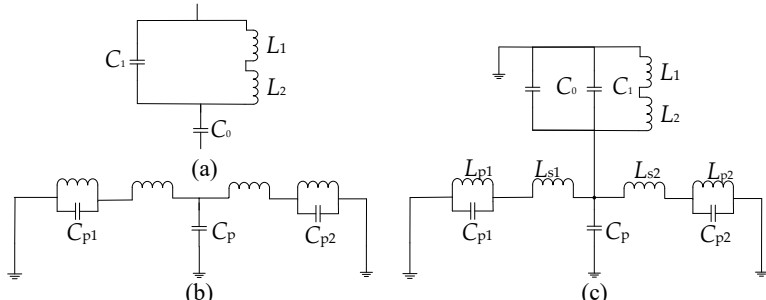

**Figure 12.** (**a**) The equivalent circuit of Archimedean spiral electromagnetic bandgap structure (ASEBG) structure resonant unit; (**b**) the equivalent circuit of the dual notch bandpass Ultra Wideband (UWB) filter; (**c**) the equivalent circuit of the proposed triple notches bandstop microstrip filter.

The structural parameters of the proposed filter are extracted by ADS as shown in Table 2. Figure 13 shows the comparison of the simulation results by using HFSS and ADS. The simulation results of the resonant frequencies of the three stopbands are identical at 3.5, 5.2, and 7.4 GHz, which further verifies the correctness and the validity of the equivalent circuit of the Archimedean spiral EBG structure and the equivalent circuit of the proposed triple notches bandstop microstrip filter. The function of notch resonator can be realized by coupling Archimedean spiral EBG structure, in which the resonance performance can be adjusted by changing the parameters of Archimedean spiral electromagnetic bandgap structure (ASEBG).

**Table 2.** The extracted parameters in Figure 10c by using ADS.

| *C* (pF) | *C* (pF) | *L* (nH) | *L* (nH) |
|---|---|---|---|
| $C_{p1} = 1.0675$ | $C_0 = 2.004$ | $L_1 = 2.48$ | $L_{p1} = 0.871$ |
| $C_p = 0.06$ | $C_1 = 1.200$ | $L_2 = 5.07$ | $L_{s1} = 0.378$ |
| $C_{p2} = 0.9803$ | - | $L_{p2} = 0.4688$ | $L_{s2} = 2.189$ |

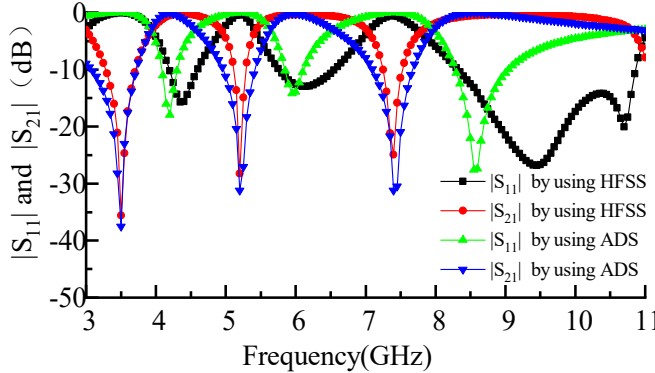

**Figure 13.** The comparison of simulation results by using HFSS and ADS.

## 4. Results and Discussion

In this paper, a tri notch stopband ultra-wideband filter is realized by coupling Archimedean spiral EBG structure on the basis of designed ultra-wideband dual-notch microstrip filter [9]. Figure 14a

shows a photograph of the proposed filter. Figure 14b shows the results of comparison between simulated data and measured data. As shown in Figure 14b, the filter has three notch resonant center frequencies of 3.5, 5.2, and 7.4 GHz. The 3 dB bandwidths of the proposed filter are 1.2, 0.65, and 0.83 GHz, and the maximum insertion loss of stopband of the filter is 33.6, 24.8, and 21.7 dB by using Vector Network Analyzer to measure the filter performance, respectively.

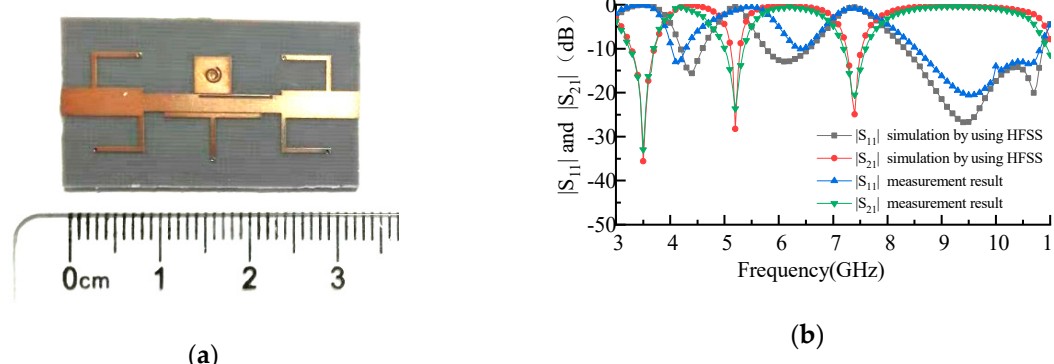

**(a)**

**(b)**

**Figure 14.** (**a**) The photograph of the fabricated triple notches bandstop microstrip filter using ASEBG; (**b**) the simulation results of the filter by using HFSS.

According to the literature [9], changing $L_5$ can adjust the resonance characteristics of the first stopband produced by T-shaped resonators. Adjusting $L_7$ can change the resonance characteristics of the third stopband produced by L-shaped defective microstrip structure. In order to realize the notch in the 5.2 GHz band, the second stopband is generated by coupling Archimedean spiral EBG structure. The material of the dielectric substrate is RT/Duorid5880 (Rogers Corporation, Chandler, AZ, USA) and the thickness is 1 mm. The final size of the filter is optimized by 0.01 GHz stepped scan frequency using HFSS. The optimized filter structural parameters by scanning frequency analysis of parameters ($d$, $W$, $r$, $dr$, $start\_t$, $end\_t$) using HFSSare shown in Table 3.

**Table 3.** Optimized structural parameters of the proposed filter (the units of length are mm, the units of angle are radian).

| Length | Length | Width | Width | Radius | Gap Length |
|--------|--------|-------|-------|--------|------------|
| $L_1 = 15$ | $L_5 = 5.3$ | $W_1 = 1.95$ | $W_5 = 0.5$ | $R_1 = 0.15$ | $g_1 = 0.1$ |
| $L_2 = 5$ | $L_6 = 10.6$ | $W_2 = 3$ | $W_6 = 0.3$ | $R_2 = 0.15$ | $r = 0.3$ |
| $L_3 = 4$ | $L_7 = 9$ | $W_3 = 0.3$ | $W_7 = 0.1$ | $W = 4$ | $a = 0.06$ |
| $L_3 = 4$ | $L_8 = 0.1$ | $W_4 = 1.4$ | $W_8 = 0.3$ | $d = 0.1$ | $d_r = 0.2$ |
| $d_{in} = 0.6$ | $d_{out} = 1$ | $W_{11} = 4$ | $W_{12} = 7.7$ | $start\_t = 0$ | $end\_t = 9$ |

*4.1. First Stopband Analysis*

The resonant frequency of the first stopband $f_1$ can be adjusted by changing the $L_5$ and $W_5$ of the T-shaped resonator while keeping other parameters unchanged and $L_6 = 2 L_5$. As shown in Figure 15a,b, $f_1$ decreases with the increase of $L_5$. However, $f_1$ increases with the increase of $W_5$. In addition, with the increase of $L_5$, the central frequencies of the second stop band $f_2 = 5.2$ GHz and the third stop band $f_3 = 7.4$ GHz, both of which do not change with $L_5$ and $W_5$ of the T-shaped resonator. Therefore, the first stopband resonance frequency of the tri notch stopband filter can be adjusted by adjusting $L_5$ and $W_5$ of the T-shaped resonator.

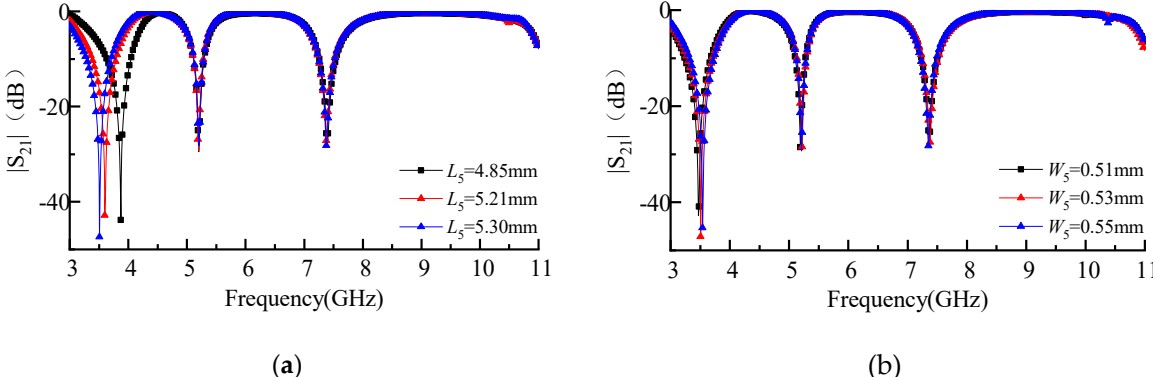

**Figure 15.** (**a**) Simulation results with different $L_5$ ($L_5$ = 4.85 mm, $f_1$ = 3.87 GHz, $L_5$ = 5.21 mm, $f_1$ = 3.6 GHz, $L_5$ = 5.3 mm, $f_1$ = 3.51 GHz); (**b**) simulation results with different $W_5$ ($W_5$ = 0.51 mm, $f_1$ = 3.47 GHz, $W_5$ = 0.51 mm, $f_1$ = 3.5 GHz, $W_5$ = 0.55 mm, $f_1$ = 3.54 GHz).

### 4.2. Second Stopband Analysis

According to the analysis in the previous section, the resonant characteristics of the second stopband can be adjusted by adjusting the parameters $W_{11}$, $r_{11}$, $dr$, $d$ $start\_t$, and $end\_t$ of Archimedean spiral EBG structure. The above parameters are analyzed by sweeping frequency while keeping other parameters unchanged in turn.

As shown in Figure 16b–d,f with the increase of $W_{11}$, $r_{11}$, $dr$, and $end\_t$, $f_2$ decreases gradually. As shown in Figure 16a,e, with the increase of $d$ and $start\_t$, $f_2$ increases gradually. As $d$ is the distance between EBG and dual notch filter, the coupling strength between EBG and double notch filter is directly affected. Therefore, when adjusting parameters of EBG, it is necessary to determine the value of $d$ first. As shown in Figure 16a–f, when the parameters $W_{11}$ $r_{11}$ $d$, $start\_t$, and $end\_t$ of EBG are changed to adjust $f_2$, $f_3$ will also change. This is because the tri notch filter is realized by coupling the EBG structure to the L-defect structure resonator. The coupling energy between the EBG and the dual notch filter changes with the structural adjustment of the EBG.

However, with the increase of $dr$, $f_2$ decreases significantly, while $f_3$ remains unchanged, which indicates that $dr$ does not affect the energy coupling between EBG and L-defect resonators. Therefore, the second notch resonance characteristics can be independently controlled by adjusting the size of $dr$ after determining the parameters $W_{11}$ $r_{11}$ $d$, $start\_t$, and $end\_t$. After the above optimization and adjustment, the parameters of the EBG structure are determined, as shown in Table 3.

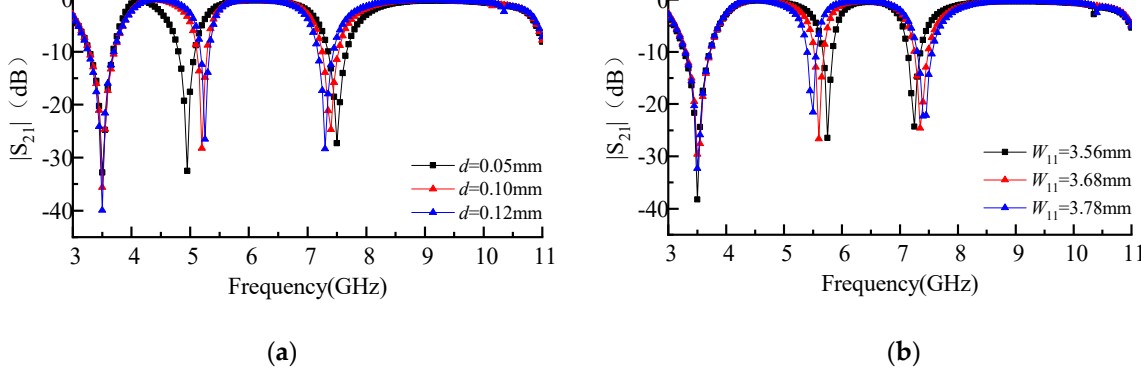

**Figure 16.** *Cont.*

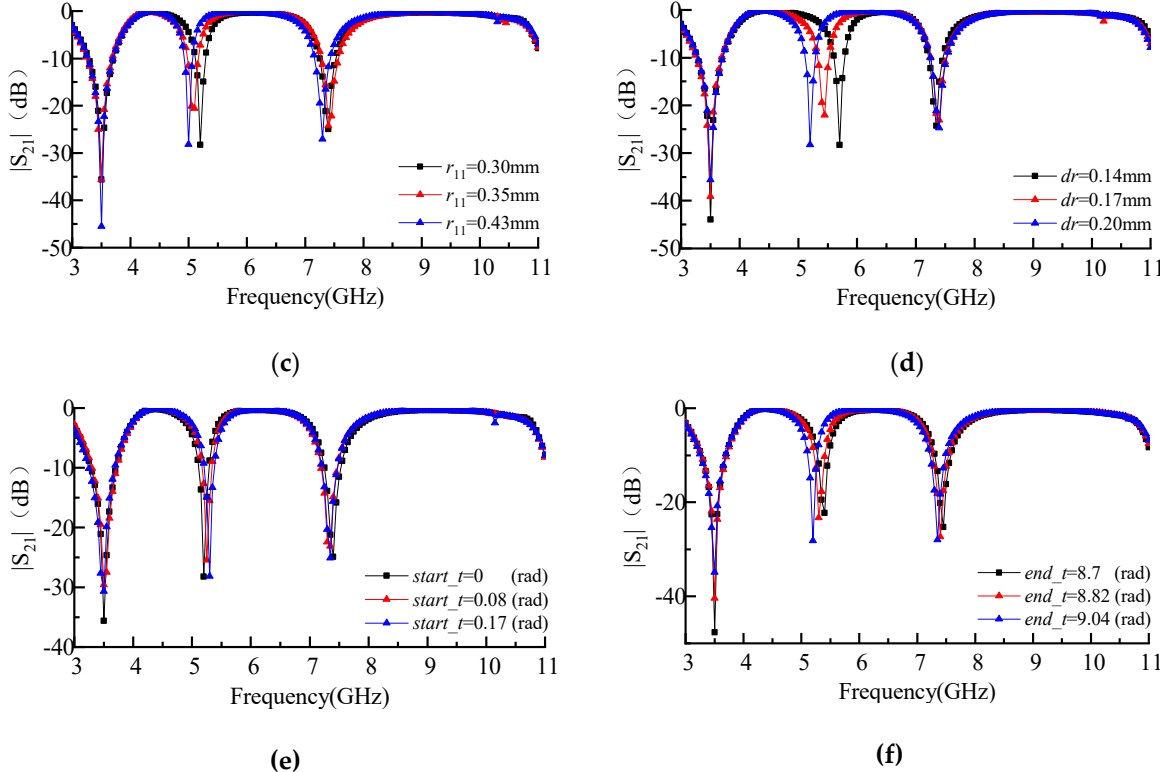

**Figure 16.** (**a**) Simulation results with different $d$ ($d = 0.05$ mm, $f_2 = 4.9$ GHz; $d = 0.10$ mm, $f_2 = 5.2$ GHz; $d = 0.12$ mm, $f_2 = 5.25$ GHz); (**b**) simulation results with different $W_{11}$ ($W_{11} = 3.56$ mm, $f_2 = 5.75$ GHz; $W_{11} = 3.68$ mm, $f_2 = 5.6$ GHz; $W_{11} = 3.78$ mm, $f_2 = 5.5$ GHz); (**c**) simulation results with different $r_{11}$ ($r_{11} = 0.3$ mm, $f_2 = 5.2$ GHz; $r_{11} = 0.35$ mm, $f_2 = 5.1$ GHz; $r_{11} = 0.43$ mm, $f_2 = 5.0$ GHz); (**d**) simulation results with different $dr$ ($dr = 0.14$ mm, $f_2 = 5.7$ GHz; $dr = 0.17$ mm, $f_2 = 5.45$ GHz; $dr = 0.2$ mm, $f_2 = 5.2$ GHz); (**e**) simulation results with different *start_t* (*start_t* = 0 rad, $f_2 = 5.2$ GHz; *start_t* = 0.08 rad, $f_2 = 5.25$ GHz; *start_t* = 0.17 rad, $f_2 = 5.3$ GHz); (**f**) simulation results with different *end_t* (*end_t* = 8.7 rad, $f_2 = 5.4$ GHz; *end_t* = 8.82 rad, $f_2 = 5.3$ GHz; *end_t* = 9.04 rad, $f_2 = 5.2$ GHz).

### 4.3. Third Stopband Analysis

The resonant frequency of the third stopband $f_3$ can be adjusted by adjusting $L_7$ and $W_7$ of the L-shaped defect microstrip structure resonator while keeping other parameters unchanged. As shown in Figure 17a,b, $f_3$ decreases with the increase of $L_7$, but $f_3$ increases with the increase of $W_7$.

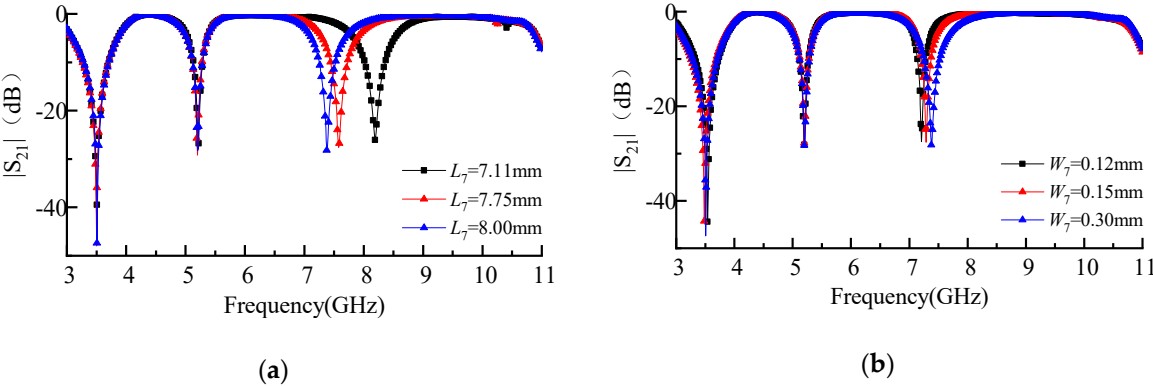

**Figure 17.** (**a**) Simulation results with different $L_7$ ($L_7 = 7.11$ mm, $f_3 = 4$ GHz; $L_7 = 7.75$ mm, $f_3 = 3.6$ GHz; $L_7 = 8$ mm, $f_3 = 3.5$ GHz); (**b**) simulation results with different $W_7$ ($W_7 = 0.12$ mm, $f_3 = 3.4$ GHz; $W_7 = 0.15$ mm, $f_3 = 3.5$ GHz; $W_7 = 0.30$ mm, $f_3 = 3.7$ GHz).

In addition, with the increase of $L_7$, the first stopband central frequency $f_1$ = 3.5 GHz and the second stopband central frequency $f_2$ = 5.2 GHz, both of which do not change with $L_7$ and $W_7$ of the L-shaped defect microstrip structure resonator. Therefore, the third stopband resonant frequency can be adjusted by adjusting $L_7$ and $W_7$.

Finally, Table 4 shows the comparison of the proposed filter with other triple notches bandstop microstrip filters [15,17,21,23–29,35]. Among them, $f_1$, $f_2$, and $f_3$ represent the central resonance frequencies of the first stopband, the second stopband and the third stopband, $MSA_1$, $MSA_2$, and $MSA_3$ represents the maximum stopband attenuation of the first stopband, the second stopband, and the third stopband, $FBW_{3dB\_1}$, $FBW_{3dB\_2}$, and $FBW_{3dB\_3}$ represent 3dB fractional bandwidth of the first stopband, the second stopband, and the third stopband. $FBW_T$ represents total fractional bandwidth of the UWB filter. The simulation results show that the three stopband center frequencies of the proposed filter are 3.5, 5.2, and 7.4 GHz, and 3 dB Bandwidth is 1.07, 0.41, and 0.61 GHz, respectively. The 3 dB FBW of the filter is 15.62%, 7.81%, and 8.91% respectively. The maximum stopband attenuation is 35.61, 28.23, 24.90 dB, and the total fractional bandwidth of the UWB filter is 112.8%.

**Table 4.** Comparison of the proposed filter with other triple notches bandstop microstrip filters.

| Ref. | $f_1/f_2/f_3$ (GHz) | $MSA_1/MSA_2/MSA_3$ (dB) | $FBW_{3dB\_1}/FBW_{3dB\_2}/$ $FBW_{3dB\_3}$ (%) |
|---|---|---|---|
| [15] | 2.4/3.5/5.2 | 15/13/20 | 4.2/2.9/4.5 |
| [17] | 5.2/5.85/8 | 20.1/25.6/27.7 | 14.4/5.2/10.1 |
| [21] | 5.2/5.8/8.0 | 20.1/23.2/24.6 | 1.8/2.3/2.1 |
| [23] | 3.6/5.9/8 | 15.9/25.6/27.3 | 2.9/3.7/2.3 |
| [24] | 2.4/3.5/5.25 | 16.5/18/14.5 | 5/3.7/4.2 |
| [25] | 1.4/2.4/3.4 | 51/44/37 | 40.1/23.1/50.2 |
| [26] | 1.56/2.45/3.46 | 40.2/35.1/33.2 | 13.6/8.6/8.1 |
| [28] | 6.1/6.9/7.6 | 15/14/13 | 2.3/5.2/1.7 |
| [29] | 5.4/5.8/8.2 | 18.3/21.2/21.5 | 13.4/6.4/5.1 |
| [35] | 2.4/3.5/5.25 | 14.3/15/16.8 | 6.2/12.2/11.8 |
| This work | 3.5/5.2/7.4 | 35.61/28.23/24.90 | 15.62/7.81/8.91 |

## 5. Conclusions

In this paper, a novel triple notches bandstop microstrip filter is proposed, which is based on the realized dual stopband filter and Archimedean spiral EBG structure resonator. The three stopband resonant frequencies of the proposed filter are 3.5, 5.2, and 7.4 GHz respectively, and the corresponding maximum stopband attenuation of them are 35.61, 28.23, 24.90 dB, respectively. The consistency of simulation results by using HFSS and ADS verifies the correctness of equivalent circuit model of Archimedean spiral EBG structure and the equivalent resonant circuit model of proposed filter. Therefore, Archimedean spiral EBG structure can be used to realize the notch function of the microstrip filter, and the proposed filter can effectively avoid the interference of wireless communication system in WIMAX band (3.5 GHz), WLAN band (5.2 GHz), and C band of satellite downlink (7.4 GHz).

**Author Contributions:** X.Z. and T.J. conceived and designed the experiments; X.Z. and T.J. analyzed the data; X.Z. wrote the paper.

**Funding:** This paper was funded by the International Exchange Program of Harbin Engineering University for Innovation-oriented Talents Cultivation.

**Acknowledgments:** This work was partially supported by the National Key Research and Development Program of China (2016YFE0111100), Key Research and Development Program of Heilongjiang (GX17A016), the Science and Technology innovative Talents Foundation of Harbin (2016RAXXJ044), the Natural Science Foundation of Beijing (4182077) and China Postdoctoral Science Foundation (2017M620918).

**Conflicts of Interest:** The authors declare no conflict of interest. The funders had no role in the design of the study; in the collection, analyses, or interpretation of data; in the writing of the manuscript, or in the decision to publish the results.

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
