# Peer review of "Triple Notches Bandstop Microstrip Filter Based on Archimedean Spiral Electromagnetic Bandgap Structure"

_electronics, doi:10.3390/electronics8090964_

Round 1

Reviewer 1 Report

Based on on the realized dual stopband filter and Archimedean spiral EBG structure resonator, a triple notches bandstop microstrip filter is proposed in this research.

I find that  the abstract include the important points of the paper, but it is to long  und it needs to be reduced.

The pictures are clear and significant enough.

The conclusion reflects the idea of this work.

 How can you explain us the correct physical effect of the spiral EBG and how what is the advantage of them opposite to oder simple slot topologies.

I will be interesting if the author shows us  the development of the filter results deponding of the each step of the filter topology.

I miss the fabrication and its corresponding measaurement results. 

I miss the comparison between the measurement and simulation results.

Did the author take into consideration the effect of the loss about the results while this work?

I would like to understand why the author didn't show us the result of the BSF along the range from DC to 3 GHz

 Can you show us the method, which you used in order to calculate the dimensions of the Slot celle?

I think authors should add this reference and discuss it in the references:

A. Boutejdar, S.D. Bennani, Design and Fabrication of Tri-Stopband Bandstop Filters Using Cascaded and Multi-Armed Methods, Advanced Electromagnetics AEM 6 (3), 18-24, 2017.

A. Boutejdar, A. Omar, Miniaturized lowpass and bandstop filters using controlled coupling of open‐loop‐ring defected ground structure, Microwave and Optical Technology Letters 52 (11), 2575-2578, 2010.

Ahmed Boutejdar; Soumia Elhani; Saad Dosse Bennani, Design of a novel slotted bandpass-bandstop filters using U-resonator and suspended multilayer-technique for L/X-band and Wlan/WiMax applications,  2017 International Conference on Electrical and Information Technologies (ICEIT), pp: 1-7, 2017

A Boutejdar et. al., A miniature 5.2‐GHz bandstop microstrip filter using multilayer‐technique and coupled octagonal defected ground structure, Microwave and Optical Technology Letters 51 (12), 2810-2813, 2009.

The paper needs major corrections

Author Response

Point 1: I find that the abstract include the important points of the paper, but it is to long und it needs to be reduced.

Response 1: We revised the abstract and streamlined its contents. We have revised the content of the paper in lines 12-23.

Abstract: With the development of artificial electromagnetic structures, defective ground structure (DGS), defective microstrip structure (DMS) and electromagnetic bandgap (EBG) are widely used in the design of microstrip filters. In this paper, triple notches ultra wideband bandstop microstrip filter based on Archimedean spiral EBG structure is proposed. The equivalent circuit of Archimedes spiral EBG is analyzed, and L and C values are extracted by ADS. The correctness of the lumped parameter model is verified by comparing the simulation results of HFSS and ADS. The influence of Archimedes spiral EBG structure parameters on resonance is analyzed by HFSS simulation. The Archimedes spiral EBG structure is coupled to the existing double notch microstrip filter to implement the triple notches filter, and the third notch is introduced by coupling mode,which can be used in the design of other microstrip devices that need notch band. The central frequencies of the three band gaps of the proposed triple notch microstrip filter are 3.5 GHz, 5.2 GHz and 7.4 GHz, and the corresponding maximum attenuation of the three stopbands are 35.61 dB, 28.23 dB and 24.90 dB, respectively.

Point 2: How can you explain us the correct physical effect of the spiral EBG and how what is the advantage of them opposite to oder simple slot topologies.

Response 2: We add the traditional MEBG structure and its parameter description. By comparing MEBG with ASEBG, it further shows that ASEBG can realize the resonator characteristics more flexibly in 211-215 line.

Point 3: I will be interesting if the author shows us the development of the filter results deponding of the each step of the filter topology.

Response 3: We analyze the traditional MEBG structure and the parameters of its equivalent circuit in line 65-119, and further illustrate that ASEBG (Archimedes spiral defect microstrip structure) can introduce a new spiral inductor L2 into the equivalent circuit of MEBG, and optimize the characteristics of the resonator by optimizing the parameters of the spiral.

Point 4:  I miss the fabrication and its corresponding measaurement results.

I miss the comparison between the measurement and simulation results.

Response 4: In this paper, we supplement the physical photos and test results of the filter, and compare the simulation results with the test results. The results further verify the correctness of the proposed spiral EBG equivalent circuit in line 241-261.

Point 6: I would like to understand why the author didn't show us the result of the BSF along the range from DC to 3 GHz. 

Response 6: The triple notches ultra wideband bandstop microstrip filter designed in this paper are all working in UWB band, ranging from 3.1GHz to 10.6GHz, so the simulated frequency is set to 3-11GHz.

Point 7:  Can you show us the method, which you used in order to calculate the dimensions of the Slot celle?

Response 7: By comparing MEBG with ASEBG, the inductive L2 produced by ASEBG introducing helix is illustrated.

Point 8:  I think authors should add this reference and discuss it in the references.

Response 8: In the introduction section of this article, we quote the literature you suggested in 50-52 line.

using cascaded and multi-armed methods [32], using controlled coupling of open‐loop‐ring defected ground structure[33], using U-resonator and suspended multilayer-technique [34], using multilayer technique and coupled octagonal defected ground structure[35]. 

Reviewer 2 Report

In this paper, another EBG structure is presented and simulated by means of ADS and HFSS. 

The paper is well written. However, experimental measurement results are missing and it must be included.

Author Response

Point 1:  The paper is well written. However, experimental measurement results are missing and it must be included.

Response 1:  In this paper, we supplement the physical photos and test results of the filter, and compare the simulation results with the test results. The results further verify the correctness of the proposed spiral EBG equivalent circuit in line 241-261.
